# Low Power Contactless Bioimpedance Sensor for Monitoring Breathing Activity [note 1]

**DOI:** 10.3390/s21062081

**Published:** 2021-03-16

**Authors:** Marko Pavlin, Franc Novak, Gregor Papa

**Affiliations:** Computer Systems Department, Jožef Stefan Institute, 1000 Ljubljana, Slovenia; marko@pavlin.si (M.P.); franc.novak@ijs.si (F.N.)

**Keywords:** bioimpedance sensor, resonator, permittivity measurement, low power

## Abstract

An electronic circuit for contactless detection of impedance changes in a tissue is presented. It operates on the principle of resonant frequency change of the resonator having the observed tissue as a dielectric. The operating frequency reflects the tissue dielectric properties (i.e., the tissue composition and on the tissue physiological changes). The sensor operation was tested within a medical application by measuring the breathing of a patient, which was an easy detectable physiological process. The advantage over conventional contact bioimpedance measurement methods is that no direct contact between the resonator and the body is required. Furthermore, the sensor’s wide operating range, ability to adapt to a broad range of measured materials, fast response, low power consumption, and small outline dimensions enables applications not only in the medical sector, but also in other domains. This can be extended, for example, to food industry or production maintenance, where the observed phenomena are reflected in dynamic dielectric properties of the observed object or material.

## 1. Introduction

Therapeutic decisions in the intensive care unit are based on measurement results from different sensors indicating the status of a patient’s vital signs (body temperature, pulse rate, respiration rate, and blood pressure) and other important parameters (e.g., blood volume and flow resulting in cardiac output). Impedance pneumography is a commonly-used technique to measure respiration rate. The operation principle is based on changing impedance of the thorax. The thorax represents a volume conductor with a constant and variable component of volume impedance [1]. The cardiac output is calculated from the stroke volume measurements. The stroke volume is the difference between the end diastolic volume and the end systolic volume. Stroke volume can be measured with invasive and non-invasive methods. One of the non-invasive methods to determine cardiac output is by thoracic electrical bioimpedance measurement [2]. During measurement the AC current is injected via electrodes. The amplitude and phase shift of the voltage response across electrodes provide enough information to calculate complex impedance at specific signal frequency. Measuring the tissue impedance with such a method requires galvanic contact between the measuring equipment and the patient, which is not possible in some specific scenarios, such as heavy compromised skin or similar issues.

A proper impedance model is required to investigate the impedance variations within the thorax due to different physiological processes. There are many different models for whole human body or just thoracic volume with accurate tissue properties; model with identified eight dominant thoracic tissues [3], finite difference human thorax model [4], a matrix method [5], a parametric model [6], and a mathematical lung tissue model [7].

Standalone low power contactless bioimpedance sensors are rare, while the combination of biopotential and bioimpedance sensors is more common. They can be seen as wireless ECG with integrated respiratory functions. Integrated circuits supporting such a combination were developed and are commercially available from different vendors, with their series of ECG analog front-ends with integrated respiratory support.

Another option for investigation in this area can be oriented on the sensor, which is more intrinsic and can operate with lower power. This is presented with examples of a passive skin patch sensor [8], a passive sensor based on an inductive–capacitive resonant circuit [9], elastic conductive silicone-based electro-resistive bands [10], multi-material fibers arranged in the form of spiral antenna [11], an ultrasonic active source and transducer [12], and an ultrasonic-based non-invasive system for simultaneous monitoring of normal and abnormal breathing activity [13]. Microwaves are used to measure dielectric properties of the materials. The permeability perturbations are known in biological tissue caused by (patho-)physiological processes. Ring resonators are simple and efficient components for measuring electric permittivity and magnetic permeability. They were used as ring resonators for frequency stabilization [14], an ultra-low power CMOS (Complementary metal–oxide–semiconductor) oscillator using film bulk acoustic resonator [15], permittivity sensor for the dielectric characterization of liquid materials [16], metamaterial sensor for nondestructive evaluation of dielectric substrates [17], and plasmonic refractive sensor that guarantees effective coupling between the waveguide and resonator [18]. They can be realized in microstrip technology. Our approach, presented in this paper, employs such a ring oscillator structure implemented in a microstrip technology, which is used to measure the shift of the resonator frequency due to dielectric changes of the target tissue. The resulting sensor is a non-invasive device suitable for respiratory rate measurements. The approach might be especially useful in the treatment of severely burned patients. The salient feature of the proposed solution is its low-power standalone performance. It can be implemented as a lightweight portable device, which may prove particularly useful during a transfer of a patient. Furthermore, wireless communication replaces the conventional cable connection to external instruments and increases flexibility of medical treatment.

The sensor was originally conceived as a part of a PhD thesis, and the basic idea of the proposed approach is presented in [19]. In the frame of the TETRACOM coordination action [20], a demonstrator prototype was implemented as a part of a feasibility study for possible manufacturing. Encouraging results have fostered further activities, including risk management concept and conformance with IEC (International Electrotechnical Commission) 62366 Standard. While the latter activities are not in the scope of this paper, we focus in this paper primarily on the sensor design. The main findings and operating principle were also used in the development of the sensor for the hygiene control system [21]. In addition, the presented sensor solution can be applied also outside the medical sector, in domains where the observed phenomena are reflected in changing dielectric properties of the observed object, such as in the food industry when performing the quality control of the fresh food [22,23], precision agriculture and environmental monitoring [24], or in the production maintenance to inspect the quality of oils and lubricants in machineries [25].

Comparing our approach to other related cases, one can see that in [11] the focus is on antenna only, while we present the whole measurement system. The solutions proposed in [12,13] are based on ultrasound, but they are not wearable. Furthermore, in our approach, additional effort was spent in optimizing the resonator itself. The emphasis was also on miniaturizing of the sensor system to make it wearable while keeping its cost low and with low power consumption. As the sensor might be integrated into smart watches, it makes our solution also applicable to a wide range of wearable solutions, such as those related to fitness monitoring and wellbeing/self-care services.

We briefly summarize the concept and describe detailed operation principles of individual system components in Section 2. In Section 3, we present measurement results that confirm the validity of the approach. In Section 4, we address power consumption issues. Some concluding remarks are given in Section 5.

## 2. Microwave Bioimpedance Sensor

Bioimpedance signals are generated within tissue when excited by AC currents. Different physiological phenomena can be detected by measuring and observing the variability of spatial impedance. The measurement is conventionally performed with the four-point method. Impedance changes are a result of a tissue composition [26], blood volume and distribution [27], endocrine activity [28], sympathetic nervous system activity [29], and other phenomena influencing body impedance. Likewise, the measurement of chest impedance is a popular technique for monitoring respiration activity.

The low-power contactless bioimpedance circuit, described below, is used for measuring changes in dielectric properties of the observed tissue, and represents a viable alternative to the conventional bioimpedance measurement techniques. The circuit measures resonant frequency deviations in the microwave resonator. The resonant frequency depends on the permeability of the material above the microstrip planar resonator. The concept was preliminary evaluated by numerical simulations in a commercial finite-element code (Comsol Multiphysics) [30]. The model of the resonator structure was studied under different boundary conditions and loads (i.e., different human body tissues) according to the expected application in practice. Obtained results were within the range of the human bioimpedance frequency response. Simulation results were compared with the measurements on a physical prototype of the resonator. Resonator measurements were performed with a vector analyzer. The key parameter for oscillator design (scatter parameter S21—the forward voltage gain) was measured between 2 MHz and 6 GHz. The simulation and measurement results showed a good match at resonant frequencies. This confirmed the feasibility of the proposed approach. A block diagram of the sensor is shown in Figure 1. In the following sections, individual parts of the sensor circuit are described in more detail.

The demonstrated solution was built using the off-the-shelf components as a proof-of-concept and power consumption evaluation. Additional optimizations are required for potential industrialization, including further miniaturization and implementation within the single ASIC chip with the same functionality, but within the smaller single-chip solution, even lower power consumption, and lower cost. As shown by other authors, the resonator may also have other forms of construction, such as flexible antenna, as shown in [11], which is a future task for further investigation.

### 2.1. Resonator

Ring resonators are useful components for many sensing applications, such as treating esophageal malignancies [31] or measuring dispersion and wavelength [32]. Some typical measurement applications include an optimization of a microstrip substrate thickness [33], discontinuity and equivalent circuit parameters [34], dielectric and dispersion characterization [35,36], or broadband materials characterization [37]. The ring resonators are also standard structures to measure dielectric properties of printed wiring board materials [38].

When the total geometry of a resonator, the resonant frequency, and the resonance order *n* are known, the permittivity loading of the resonator can be determined. The basic ring resonator circuit is shown in Figure 2. It consists of two feed lines with coupling gaps and the resonator ring. The ring is powered via feed lines. The separation between the lines and the ring is provided by coupling gaps. The gaps represent loose coupling and should be optimized to provide proper oscillation within the ring structure. The structure supports waves that are integral multiples of the guided wavelength *λ* of the circumference (Equation (1)).
(1)nλ=πD,  n=1, 2, 3,…
where *n* is the mode number and *D* is the ring mean diameter.

We designed a resonator to evaluate the idea of measurement of chest permittivity. The designed structure was a conventional ring resonator. Geometry and prototype photography are shown in Figure 2.

Resonator operation in free air was simulated with Agilent ADS (Advanced Design System) and measured. Simulation results were compared with measurements. The resonator measurements were performed with Agilent RF Vector Analyzer N9923A. The scatter parameter S21 was measured between 2 MHz and 6 GHz. The simulation and measurement results show a good match at the resonant frequencies for mode *n* = 1 and higher modes. The calculated resonant frequency at mode *n* = 1 matched with the simulation at *f_1_* = 2.101 GHz. Higher mode results showed some difference. The calculated frequency at mode *n* = 2 was 4.162 GHz while the measured frequency was lower, at 4.134 GHz. Complete results are shown in Figure 3. The measurement results of S21 are quite similar to the simulation results, but due to simplified material data at higher frequencies, it will cause some matching errors at frequencies above 4 GHz. However, the area of interest at mid frequencies is still fitting the simulation results. The lower frequencies are not well visible in the Figure 3, but also the usability of the resonator is limited to regions with higher than −40 dB.

The measurement setup for S21 was as follows: Start frequency: 2 MHz, stop frequency: 6 GHz, resolution: 201 points, output power: high, interference rejection: off, system: Z0 50 Ohm, calibration: mechanical 2 port calibration.

### 2.2. Low-Pass Filter

The oscillator was implemented with BFP520 low voltage silicon NPN RF bipolar transistor (Infineon Tech., Neubiberg, Germany). As shown in Figure 1, the ring resonator is connected to the amplifier via a low-pass filter, which is used to reject all higher harmonics above the maximum operating frequency of the resonator. The selected BFP520 has a wide operating range. The reason why a filter is placed at the amplifier input is the reduction of electromagnetic noise. Instead of amplifying all harmonics and then filtering out the unwanted spectrum, the signal is filtered at the amplifier input. The complete oscillator prototype is shown in Figure 4.

### 2.3. Frequency Divider

Signals in the frequency range between 500 MHz and 3 GHz are fed to the frequency divider uPB1507GV, connected to the output of the oscillator. The following pin-selectable division factors are provided: 64, 128, and 256.

### 2.4. Frequency Synthesizer

The traditional frequency synthesizer is analogue Phase-Locked Loop (PLL). On the other hand, direct digital synthesis used in our case has some advantages over conventional PLLs. The architecture of the employed frequency synthesizer, shown in the block diagram of Figure 5, provides highly accurate signals with rapidly changeable frequency consuming only a portion of the power required by the conventional analogue PLL. The direct digital synthesizer (DDS) generator employs a phase accumulator. The output of the phase accumulator serves as the address to the lookup table, which converts linear phase (amplitude over time) to the corresponding digital amplitude of a sine wave between 0° and 360°. The output of the sine lookup table drives the high-speed digital to analogue converter (DAC). The output signal is filtered with a low-pass filter and amplified to provide accurate sine wave, with fast settling time.

The phase accumulator has *m*-bit internal register. There are 2^m^ possible phase points. The phase accumulator register is incremented each clock cycle for exactly N, also known as “DDS tuning word”. The frequency of the output sine wave V_OUT_ is equal to:(2)$$f_{\mathrm{OUT}}=\frac{{\mathrm{Nf}}_{C}}{2^{m}}$$

The DDS resolution is defined when *N* = 1:(3)$$f_{\mathrm{INC}}=\frac{f_{C}}{2^{m}}$$

The described DDS generator is extremely flexible, stable, and has high resolution. The frequency changes and settling times are minimal, which is important for cyclic operation in a low-power system. The complete frequency synthesizer was realized with a low-power 75 MHz single chip device AD9834. It has m = 28 bits phase register. At maximum clock frequency, 75 MHz, the resolution of 0.28 Hz steps can be achieved. It has a power-down feature to minimize the current consumption when not in use.

### 2.5. Frequency Mixer

The output signals from the frequency divider and DDS synthesizer are fed to the mixer. It is used to provide frequency translation. The result of the down-conversion is a low-frequency signal. The frequency of the mixer output signal is within the acceptable range for the microcontroller. The signal from the frequency divider shown in the block diagram (Figure 1) is a sine wave at frequency *f_R_*. It is mixed (multiplied) with the sine wave signal from DDS at frequency *f_D_*.

The resulting signal is filtered with a low-pass filter to remove the upper product *f_R_ + f_D_*. When *f_D_* is close to *f_R_*, the output signal from the mixer has a very low frequency. The frequency changes of the input signal from the resonator can be detected in a wide frequency range by applying proper injection signals *f_D_* to the mixer. This can provide a very high dynamic range of sensor detection.

The complete module (except oscillator) for prototyping was realized on a small board, shown in Figure 6. The architecture shown in the block diagram in Figure 1 has a wide operating range and agile frequency adjustments. Such a wide operating range provides great flexibility. On the other hand, it takes some effort to control the device within the operating range after placing the device to the patient. First, the initial search is performed to find the resonant frequency. The frequency strongly depends on the position and the permeability of the surrounding tissue. In order to place it within the operating range of the microcontroller, the signal is transformed in two steps. In the first step it is divided by the frequency divider and in the following step it is subtracted by the DDS output frequency within the frequency mixer. In other words, the DDS output is adjusted to the frequency near the detected frequency of the resonator (divided via frequency divider). The DDS tracks the input frequency in discrete steps, which keeps the mixer resulting frequency within the detecting range of the microcontroller. The maximum detection frequency is limited by the microcontroller hardware, while the minimum detection frequency is limited by the stability of the mixer. The output can span down to DC (*f* = 0), but it is a common practice to avoid such a case due to additional difficulties that might occur, when the frequencies of both signals are very close. In our case, the lower limit was set to 10 MHz.

## 3. Bioimpedance Sensor Testing

The sensor prototype was tested first at the component level and finally as the whole module. Long-term stability of the measured frequency was tested with the resonator placed freely in the air. The frequency response was measured on a healthy volunteer (the main author of this paper). The sampling rate was 100 samples per second. The ring resonator was placed on the chest and the initial frequency was read by an oscilloscope. This frequency was programmed into the DDS controlling software as the initial frequency. The microcontroller was connected to a PC via USB interface. The frequency of the DDS and the output frequency from the mixer were stored into a file used for the final evaluation of the sensor sensitivity.

### 3.1. Oscillator Test

Firstly, the amplifier was tested. The S21 of the open loop oscillator amplifier was measured at different supply voltages (Figure 7). The reasonable gain margin for the oscillator is around 3–8 dB. Higher gains would result in increased phase noise and oscillator instability, which could have a consequence of possible spurious products. Lower gains than 3 dB might cause slower start, or even prevent oscillations. The useful frequency range is indicated in green.

The oscillator loop was then closed with the resonator. The power supply voltage was set to 4 V, because the operating conditions at this supply voltage are within the expected 3–8 dB range.

The oscillator output was measured with a spectrum analyzer. The spurious product within the measured spectra from 1800 MHz to 2300 MHz was negligible. The resonator was measured isolated from the tissue (red trace in Figure 8) and close to the tissue with the average relative permittivity of about 45 (green trace in Figure 8). The oscillator output frequency without load was 2078 MHz, which is very close to the simulated and theoretically designed 2101 MHz.

The resonator was loaded with water balloons to simulate the chest tissue with controlled permittivity. The permittivity constant of water at 25 °C is around 80 at frequencies near 1.9 GHz [39]. The measured resonant frequency was 1689 MHz.

Finally, when placed at the chest surface (hence loaded with the dielectric with relative permittivity around 45 [40]), the measured resonant frequency was 1880 MHz. The frequency shift about 200 MHz represents significant differentiation between the air and the tissue (Figure 8). Once these conditions are established, the physiological activity will cause frequency deviations which reflect the physiological conditions of the patient.

### 3.2. Prototype Sensor Test

The whole sensor was tested and the results are shown in Figure 9. The sensor with a resonator was placed on the chest of a volunteer. The measurement cycle started by scanning the frequency range by incrementing the DDS tuning word, which controls the DDS output until the microcontroller detects the frequency of the sensor output signal. The microcontroller is connected to a USB port for logging the measured data. The USB communication is fast enough to perform update about 100 times per second. The measurement record consists of frequency measurement and DDS tuning word setup and serves for the calculation of the actual resonator frequency. The measurement setup diagram is shown in Figure 10.

The final test was performed by the volunteer, who was breathing normally. Special attention was placed to avoid any additional movements, except the breathing, to avoid any unintentional artefacts within the signal. The DDS was tuned to keep the mixer output frequency below 400 kHz, which was the maximum operating frequency of the low-power counter within the microcontroller. The frequency range at the frequency divider output was between 6.882 MHz and 7.025 MHz with the maximum bandwidth of 0.142 MHz. This range corresponds to the microwave oscillator frequency range from 1761.8 MHz to 1798.4 MHz, which is 36.603 MHz frequency shift induced by the breathing.

Test results clearly show detectable deviation in the resonator frequency during breathing (Figure 9). The main signal of interest, however, is not the absolute measurement of the chest permittivity or absolute frequency of the resonator, but the frequency (and optionally amplitude) of breathing. This is derived from the periodic output in frequency deviation. The paper is not concentrating on a digital signal processing, which could further improve the signal-to-noise ratio of the measured signals. Just running simple fast Fourier transformation and searching for dominant harmonic results in a single number representing respiration frequency. The selected low-power microcontroller has proven enough computing power to process this in real time with low power consumption [41].

Such a non-invasive, non-galvanic contact (i.e., no direct physical contact between sensor electrode and tissue) method represents an excellent advantage over the conventional breathing sensing methods. The presented method is one of the very limited choices, or maybe the only one, for the critically ill patients without possibility to directly contact any electrode on the skin surface (e.g., heavily burnt patients).

## 4. Power Consumption

Since the designed bioimpedance sensor is an autonomous device, the power consumption is an important issue. In this regard, operation of individual parts of the sensor were analyzed in order to track the most critical actions and improve any shortcomings.

When switched on, the DDS generator is in its initial state. The DDS output is deactivated for 0.7 µs until its registers are set to the required state that determines the DDS output frequency. The DDS generator remains idle for eight oscillator frequency cycles (i.e., 1.78 µs at 50 MHz frequency). The DDS generator is thus ready for operation after 2.5 µs. Power consumption of DDS generator depends on its output frequency. Measured power consumption for different output frequencies at *V_CC_* equal to 3.3 V is shown in Figure 11. Assessed average power consumption at *V_CC_* = 3.3 V and *I_CC_* = 3.5 mA is 12 mW.

Power consumption of the frequency divider as a function of the supply voltage is shown in Figure 12. The input frequency was 1.47 GHz and the ambient temperature was 25 °C. The divider consumes about 20 mA at *V_CC_* = 5 V, which is high and considerably affects the overall power consumption unless appropriate countermeasures are taken.

Power consumption of the frequency mixer as a function of the supply voltage is shown in Figure 13. The frequency of the input signal was 5.74 MHz and the frequency of the reference DDS generator was 5.2 MHz. Measurement was performed at the ambient temperature of 25 °C. Although lower than in the case of frequency divider, the power consumption is still relatively high and calls for optimization.

An established way to reduce power consumption is to redesign critical parts. While this requires additional efforts, we took an alternative approach by optimizing the timing of sensor operation. The resulting operation time and energy consumption of individual sensor parts are given in Table 1.

Table 2 summarizes some comparison with related work. A sensor with multi-material fibers integrated into textile [11] provides a cost-effective solution. The main advantage is textile flexibility which may be used as an alternative solution to our rigid resonator. However, the demonstrated solution [11] requires a larger area for the sensor. An ultrasonic contactless sensor for breathing monitoring [12] is the compact solution with greater sensitivity, with two disadvantages compared to our sensor: higher power consumption and fixed installation prevents the device being wearable. A similar situation is with a system for monitoring breathing activity using an ultrasonic radar [13]. The presented radar system requires fixed installation (not wearable) with higher cost and power consumption.

## 5. Conclusions

The presented sensor is based on the proven microwave measuring principle. Its operation was tested in practice by measuring the breathing, which proved to be an easy detectable physiological process. Our sensor provides a contactless, non-invasive treatment of patients, which can be especially helpful in cases where direct contact is not possible or would cause harm. Severely burned patients are a typical example. Such patients may also have severely affected respiratory tract, and monitoring of breathless touch can be extremely important for successful and faster recovery. An additional feature of the proposed solution is its low-power standalone performance. It can be implemented as a lightweight portable device and its wireless communication ability towards external instruments increases flexibility of medical treatment. In this regard, medical device usability aspects and risk management concept in conformance with IEC 62366 Standard are currently underway.

## Figures and Tables

**Figure 1 sensors-21-02081-f001:**
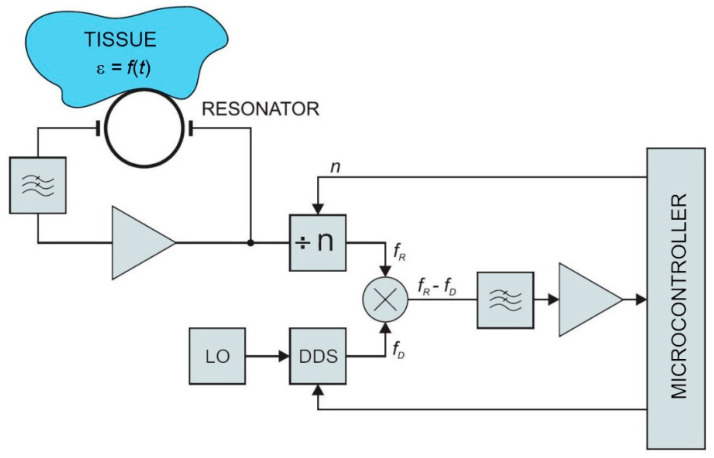
Large tuning range resonant sensor. LO is a local oscillator, DDS is a direct digital synthesizer, ÷ n is a frequency divider, ε is tissue permittivity, *n* is the mode number, *f_D_* divider frequency, and *f_R_* is resonant frequency.

**Figure 2 sensors-21-02081-f002:**
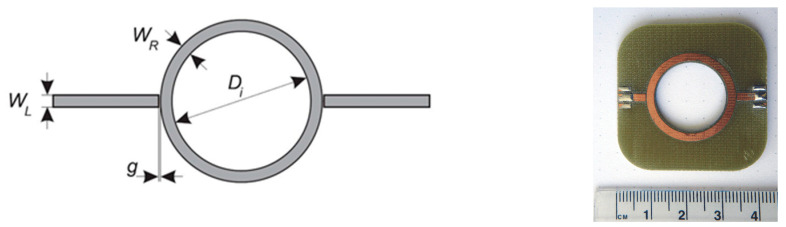
Microstrip ring resonator geometry and a prototype photo (line width *W_L_* = 2.2 mm, ring width *W_R_* = 2.2 mm, ring diameter *D_i_* = 2.2 mm, gap *g* = 0.15 mm).

**Figure 3 sensors-21-02081-f003:**
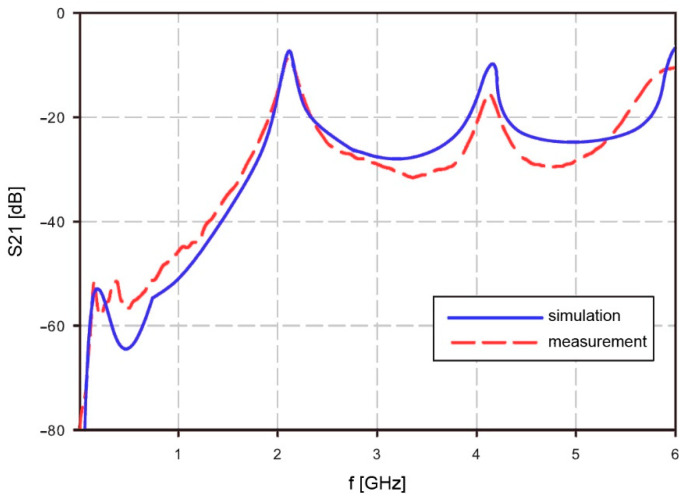
Calculated and measured S21 for resonator between 2 MHz and 6 GHz.

**Figure 4 sensors-21-02081-f004:**
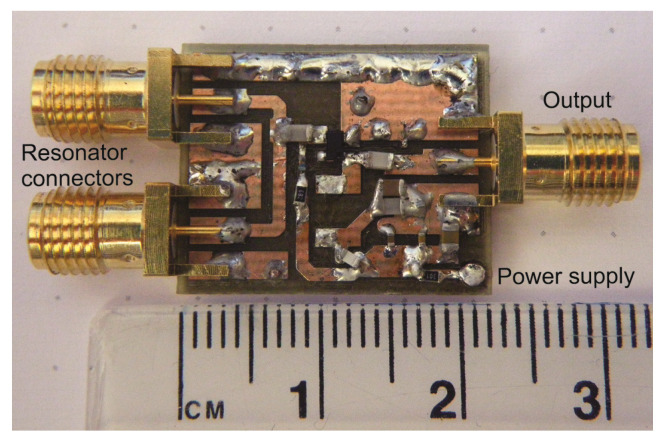
Oscillator prototype.

**Figure 5 sensors-21-02081-f005:**
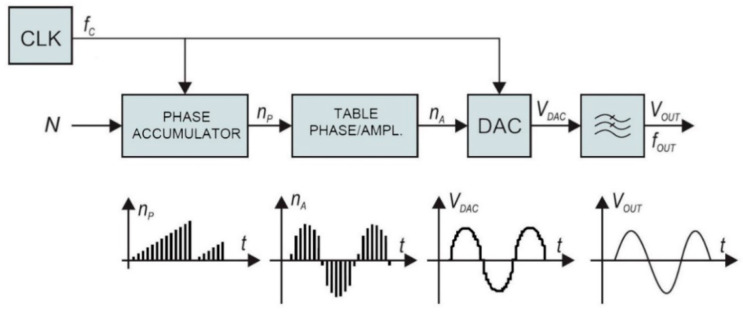
Employed frequency synthesizer.

**Figure 6 sensors-21-02081-f006:**
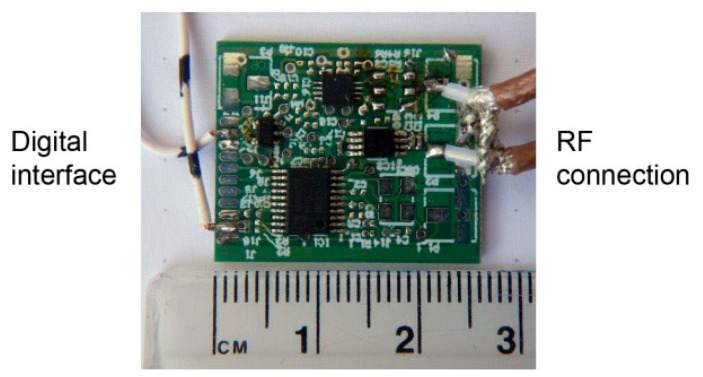
Bioimpedance sensor frontend: Frequency divider, DDS, and mixer.

**Figure 7 sensors-21-02081-f007:**
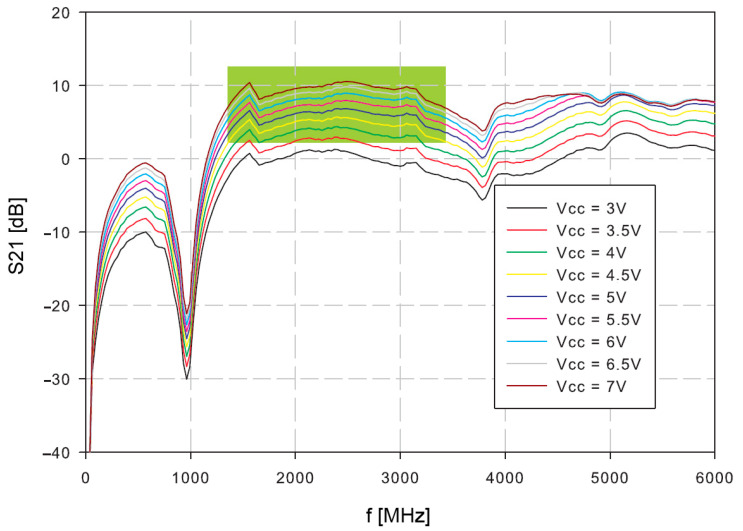
Measurement of S21 of the open-loop oscillator amplifier.

**Figure 8 sensors-21-02081-f008:**
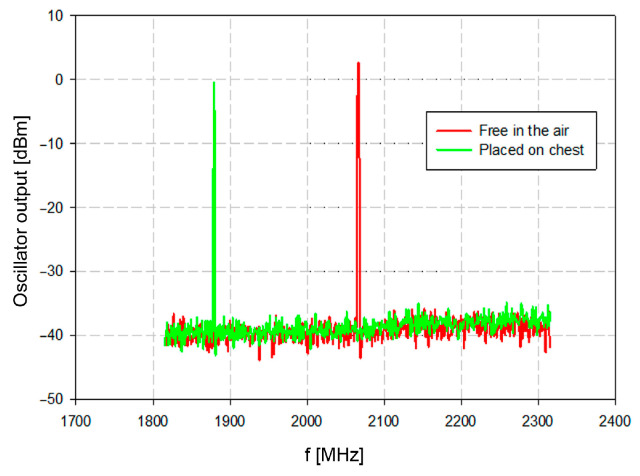
Oscillator test with two different operating conditions.

**Figure 9 sensors-21-02081-f009:**
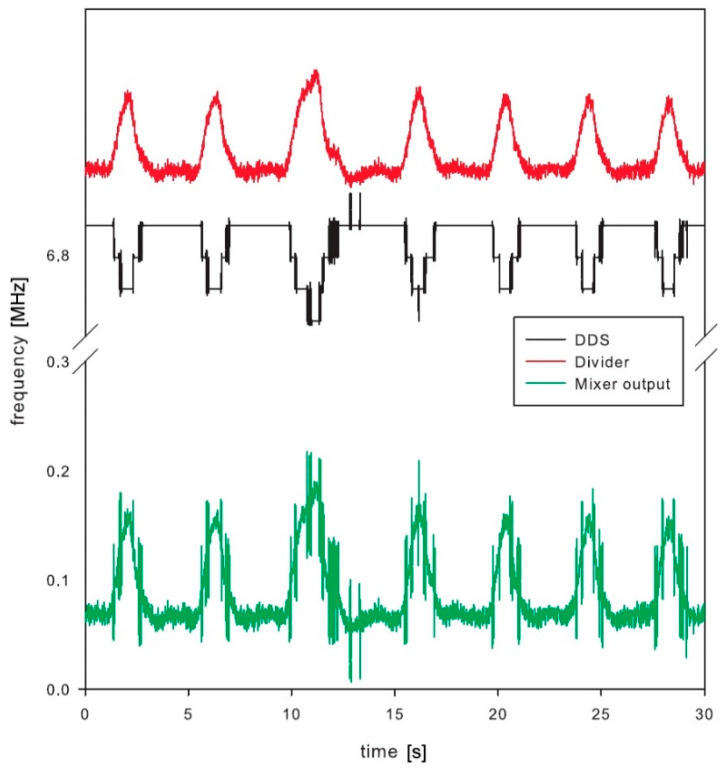
Final results from respiratory sensor test.

**Figure 10 sensors-21-02081-f010:**
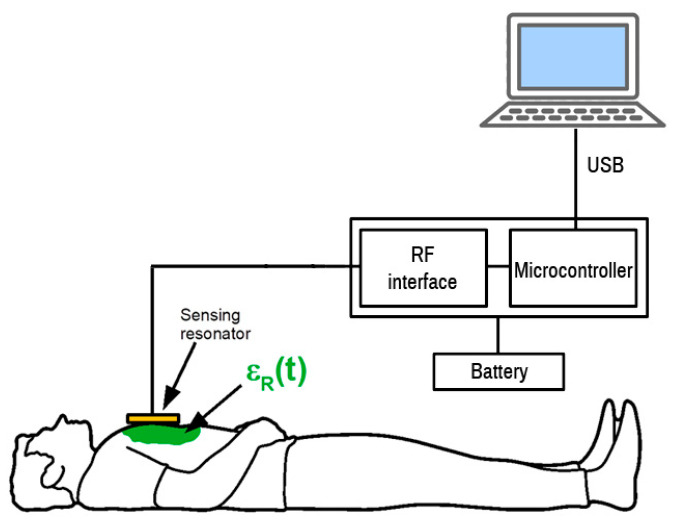
Measurement setup for the prototype test.

**Figure 11 sensors-21-02081-f011:**
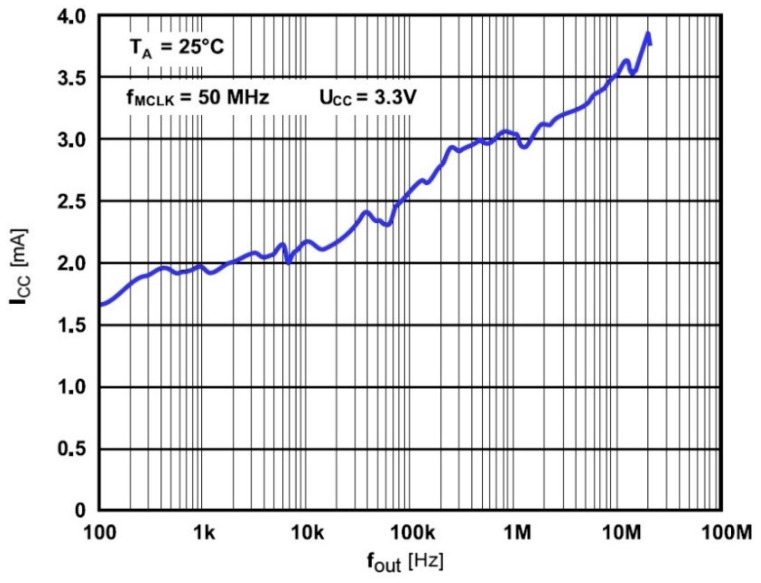
Power consumption of DDC generator as a function of the output frequency.

**Figure 12 sensors-21-02081-f012:**
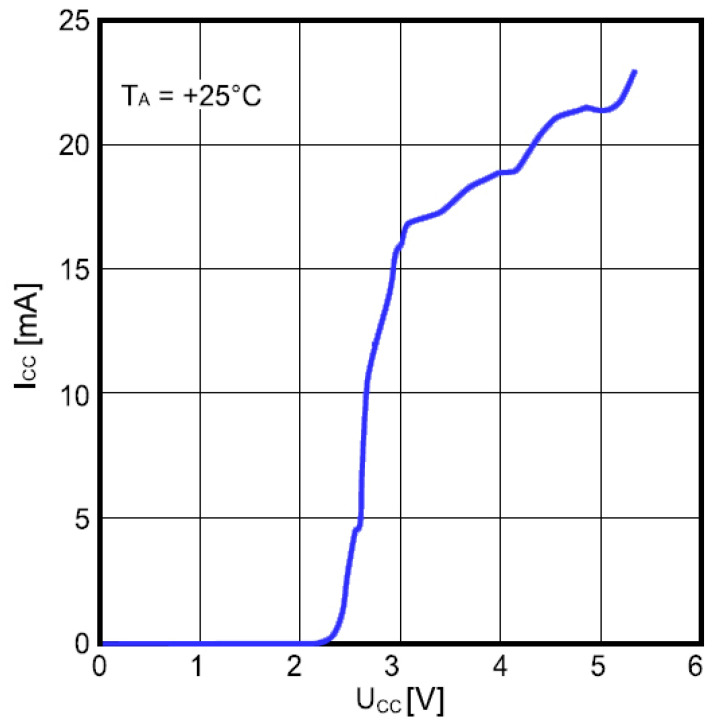
Power consumption of frequency divider as a function of the supply voltage.

**Figure 13 sensors-21-02081-f013:**
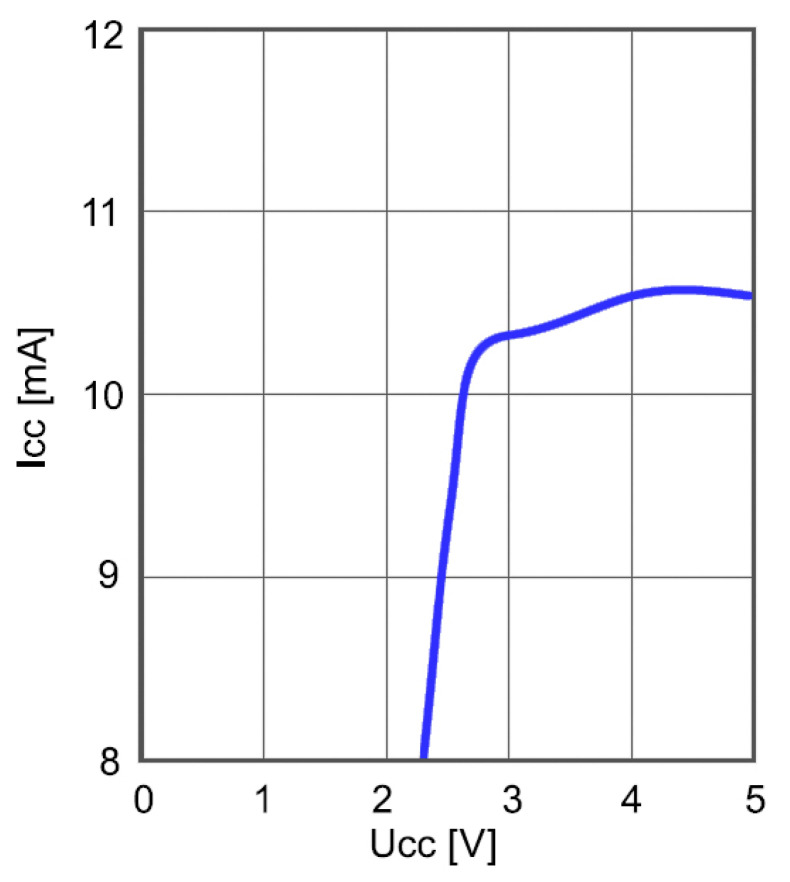
Power consumption of frequency mixer as a function of the supply voltage.

**Table 1 sensors-21-02081-t001:** Power consumption of bioimpedance sensor.

	Power (mW)	Time (µs)	Energy (µJ)
Oscillator and frequency divider	150	256.5	38.475
DDS	12	36.5	0.438
Mixer	34	34	1.156
Overall		256.5	40.069

**Table 2 sensors-21-02081-t002:** Comparison with related work.

	Sensitivity	Power Consumption	Size	Price
This work	2% (36 MHz @ 1.8 GHz)	12 mW	Wearable (sensor and readout electronics)	Low cost (approx. 15 EUR BOM cost)
[11]	0.67% (16 MHz @ 2.4 GHz)	Large (demonstrated with large instruments)	Wearable (sensor only)	Low cost (sensor antenna only)
[12]	25% 200–250 arbitrary units	Possibly low power (ultrasound transmitter and receiver)	Fixed installation (not wearable)	Mid to high
[13]	(not reported)	22.8 mW 6.936 mA × 3.3 V (assumed)	Fixed installation (not wearable)	Mid + data provider lease

## Data Availability

Not applicable.

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
