# Peer review of "Low Power Contactless Bioimpedance Sensor for Monitoring Breathing Activity"

_sensors, 2021, doi:10.3390/s21062081_

Round 1

Reviewer 1 Report

This paper additionally needs measurement setup diagram and photo to clearly explain how to measure the changing bio impedance by using the resonator.

Check the typo in line 13: having for dielectric the observed tissue ??

Author Response

Comment 1:

This paper additionally needs measurement setup diagram and photo to clearly explain how to measure the changing bio impedance by using the resonator.

Response 1:

In the revised version we provided additional measurement setup diagram as Figure 9, and more details are written on the measurement setup, to explain the sensor prototype test.

Comment 2:

Check the typo in line 13: having for dielectric the observed tissue?

Response 2:

In the revised version the particular sentence is reworded to be more clear: “… having the observed tissue as a dielectric.”

Reviewer 2 Report

Please find the reviewer's comment as an attachment.

Author Response

Comment 1:

Please emphasize the possibilities of practical application of the solution developed in the paper.

Response 1:

In the revised version the practical applications of the proposed solution are better described, emphasizing portability and wireless connectivity, which increase the flexibility of medical treatment, especially in case of transfer of patients. As well we mentioned the possible applications into fitness and wellbeing services sectors.

Comment 2:

Please emphasize the novelty of the paper (section Introduction) based on References.

Response 2:

In the revised version the novelty is emphasized through the intended use of this miniaturized sensor for severely burned patients.

Comment 3:

Please define the introduced parameters, e.g. lambda below Eq. (1).

Response 3:

In the revised paper lambda parameter is explicitly explained to determine wavelength.

Comment 4:

Table 1 ‐ please write the units of physical quantities in square brackets. The same comment applies to Figures.

Response 4:

In all Figures, Tables and texts of the revised paper the physical quantities are written in square brackets.

Comment 5:

Fig. 1. ‐ please, do not write the round brackets in italics.

Response 5:

Round brackets in Figure 1 are now corrected, not to be in italics.

Comment 6:

Below Fig. 1 ‐ the included symbols/parameters should be explained. The same comment applies to Figure 2.

Response 6:

In the revised paper included symbols and parameters are described in captions of the relevant figures.

Comment 7:

Figure 3 ‐ it is justified to determine the error between measurement and simulation.

Response 7:

In the revised version additional sentence is included that explains the source of differences between simulation and measurement at higher frequencies, caused by simplified material data.

Comment 8:

All physical quantities or parameters should be written in italics, e.g. description of Fig. 2.

Response 8:

In the revised version of the paper the physical quantities in Figure 2 are written in italics.

Comment 9:

Figure 7 ‐ please replace the comma with a dot for the voltage value. The same comment applies to the description of Fig. 2. Please check entire paper for this mistake.

Response 9:

Figure 7 and the rest of the paper was checked and correct number format is used.

Comment 10:

Please check the paper for the move of the physical quantity units by one space from the corresponding numerical value, e.g. description of Fig. 2.

Response 10:

In the revised version the physical quantity units are separated by one space from the corresponding numerical values.

Comment 11:

Please write correctly the degree sign, e.g. line 400.

Response 11:

In the revised version the degree sign was checked and written correctly.

Comment 12:

Please, do not write the pi symbol in italics, e.g. line 187.

Response 12:

In the revised paper the pi symbol is not written in italics.

Comment 13:

Please, use the same font size and font type for both the body of the paper and Figures.

Response 13:

In the revised version body text as well as all figure and table captions were checked to consist the correct font size.

Comment 14:

Please correct the paper for typos, e.g. lines: 204, 205, 206, 265, 300, …

Response 14:

In the revised manuscript several typos were corrected throughout the paper.

Reviewer 3 Report

Dear Authors.

Unfortunately, your submission to the journal cannot be accepted.

The main reason 
for this decision are:

1. Abstract and Introduction are poor.
I recommend additional/rewrite "Abstract and contribution".

2. Conclusions and Future Work are poor.

3. Completeness and Related Work are poor.

4. Poor English write-up; It is almost impossible to understand the contribution of the paper.

5. Scientific papers should be replicable.

The strength of the paper included: 
the topic is interesting.

Author Response

Comment 1:

Abstract and Introduction are poor. I recommend additional/rewrite "Abstract and contribution".

Response 1:

In the revised version we improved/extended the Abstract and Introduction, to include more details on the contribution and sensor’s wide usability.

Comment 2:

Conclusions and Future Work are poor.

Response 2:

In the revised version we extended and rewrote the section on conclusions and future work.

Comment 3:

Completeness and Related Work are poor.

Response 3:

In the revised version we provided extensions to the related work section and improved comparison Table 2. The completeness aspect is mostly related to the commercialisation of the work and is included in the future work section.

Comment 4:

Poor English write-up; It is almost impossible to understand the contribution of the paper.

Response 4:

In the revised version the English language was additionally checked and improved.

Comment 5:

Scientific papers should be replicable.

Response 5:

In the revised version several improvements to the text were introduced. Additional texts, figures, descriptions of parameters, measurement setup, and diagrams allow the reader to replicate the content of the paper.

Round 2

Reviewer 1 Report

My concerns have been resolved.

Reviewer 2 Report

All comments of the reviewer have been included in the revised version of the paper. Therefore, I recommend its publications in the present form.

Reviewer 3 Report

Dear Authors. The revision adequately address the concerns expressed in last review. So, I recommend that this revised manuscript can now be recommended for publication (Accept as is).